# Design and Synthesis of New Coumarin Hybrids Active Against Drug-Sensitive and Drug-Resistant Neuroblastoma Cells

**DOI:** 10.3390/antiox15010031

**Published:** 2025-12-24

**Authors:** Carola Grondona, Barbara Marengo, Giulia Elda Valenti, Sara Tirendi, Eleonora Russo, Cinzia Domenicotti, Bruno Tasso

**Affiliations:** 1Department of Pharmacy, University of Genoa, Viale Benedetto XV 3, I-16132 Genoa, Italy; carola.grondona@edu.unige.it (C.G.); eleonora.russo@unige.it (E.R.); 2Department of Experimental Medicine, University of Genoa, Via Leon Battista Alberti 2, I-16132 Genoa, Italy; barbara.marengo@unige.it (B.M.); giuliaelda.valenti@edu.unige.it (G.E.V.); sara.tirendi@edu.unige.it (S.T.); 3IRCCS Ospedale Policlinico San Martino, I-16132 Genoa, Italy; 4Inter-University Center for the Promotion of the 3Rs Principles in Teaching & Research (Centro 3R), I-56122 Pisa, Italy

**Keywords:** antioxidant scaffolds, Coumarin, molecular hybridization, Neuroblastoma

## Abstract

High-risk neuroblastoma (NB) is an aggressive pediatric tumor characterized by pronounced biological heterogeneity and frequent development of chemoresistance, which critically limits therapeutic efficacy. Identifying novel anti-NB agents remains an urgent unmet need. To address this, we designed and synthesized 17 hybrid molecules by combining natural antioxidant scaffolds (coumarin, vanillin, and isovanillin) through an acyl-hydrazone linker. Several derivatives significantly reduced the viability of MYCN-amplified NB cells (HTLA-230) and their multi-drug resistant counterpart (ER) while not affecting human keratinocytes (HaCat). Among them, compounds **5**, **9** and **12** selectively inhibited HTLA and ER growth (10–25%) without affecting HaCat, accompanied by robust ROS overproduction, particularly by **9** and **12** (up to 40%). None of these compounds induced apoptosis or ferroptosis. Instead, their antiproliferative effects were associated with senescence induction and, only for compound **5**, with a decrease in clonogenic potential. Moreover, to further characterize compounds **5**, **9**, and **12**, the analysis was extended across other human neuroblastoma cell lines. In parallel, the effects of the compounds on non-malignant cell lines were assessed to obtain an indication of their selectivity toward tumor cells. Compound **17**, a structural analog lacking the second aromatic ring in the ex-aldehyde portion, displayed a distinct profile with a limited anticancer activity, underscoring the importance of this structural fragment for antiproliferative efficacy.

## 1. Introduction

### Neuroblastoma

Neuroblastoma (NB) is a neural-crest-derived embryonal malignancy that almost exclusively occurs in early childhood. Despite being a relatively rare disease affecting 1 in 8000 live births, and representing 6–10% of all childhood tumors, NB accounts for 12–15% of all childhood cancer-related death and is the most common and deadly extracranial tumor in children [1].

NB shows a highly heterogenous clinical behavior, ranging from tumors curable by surgery alone or in combination with limited chemotherapy, over patients with highly metastasized disease, to the special subcategory of NB that is metastatic at diagnosis but shows spontaneous regression with no therapeutic intervention [2,3].

The clinical heterogeneity of NB is accompanied by distinct genetic patterns, including high-level MYCN amplification—seen in approximately 20–25% of cases and strongly associated with poor prognosis—and recurrent segmental chromosomal imbalances, which also correlate with aggressive disease, even in the absence of MYCN amplification [4].

The biologic heterogeneity of neuroplastic tumors that occur during childhood has resulted in a dichotomization in therapeutic strategies according to risk classification [5]. For tumors that have favorable biologic features, the clear trend has been to reduce therapeutic intensity. In contrast, the approach to tumors with adverse prognostic features has shifted over the past two decades toward intensifying chemoradiotherapy. In high-risk patients, therapy typically involves an intensive induction of the remission chemotherapy regimen—commonly referred to as COJEC—which consists of the cyclic administration of the chemotherapeutics Cisplatin, Vincristine, Etoposide, Cyclophosphamide and Carboplatin. Induction follows the consolidation of the remission phase, characterized by surgical resection of the primary tumor and myeloablative therapy in combination with autologous hematopoietic stem cell reinfusion and localized radiotherapy. Maintenance therapy comprises the administration of retinoic acid to promote differentiation of residual tumor cells, alongside immunotherapy for the eradication of minimal residual disease. Despite this, the survival outcomes remain markedly different across risk groups: low-risk patients have a 95% chance of disease-free survival, whereas high-risk patients have a 50% risk for disease relapse due to the development of chemotherapy-resistant cells [6]. Even with intensive therapy, the majority of high-risk patients eventually develop progressive disease that is refractory to further therapy [7].

Cancer cells may either exhibit a significant primary resistance to drugs (primary resistance), or acquire characteristics of multi-drug resistance after long-term chemotherapy (acquired resistance) due to mutations in genes coding for proteins involved in cell death and survival and due to epigenetic changes. NB exhibits inter- and intra-tumor genetic heterogeneity characterized by abnormal telomere maintenance mechanisms, MYCN amplification, and mutations in the RAS and/or p53 pathways, resulting in poor prognosis [8].

Efforts to identify efficacious treatments for high-risk chemotherapy-resistant NB remains a critical unmet medical need. In this context, molecular hybridization is a technique that has recently been widely used in medicinal chemistry for drug discovery and consists of joining structural motifs from two or more bioactive entities into a single molecular framework capable of reproducing the original properties of the prototype molecules or, in some cases, acting synergistically on the same or different biochemical pathways related to cancer [9].

Since alterations in the cellular redox balance caused by the generation of free radicals are frequently observed in NB, our aim was to design and synthesize novel antiproliferative substances by molecular hybridization of natural scaffolds endowed with intrinsic antioxidant activity (coumarin, vanillin, and isovanillin). These hybrids were intended to exert both antiproliferative effects, particularly against drug-resistant NB, and modulatory effects on oxidative stress pathways.

Coumarin (2H-chromen-2-one or 2H-1-benzopyran-2-one), bicyclic heterocycle consisting of benzene and 2-pyrone rings, was chosen as bicyclic part of our newly designed molecules. Coumarins are part of flavonoid group of plant secondary metabolite, and their derivatives are a wide class of natural and synthetic compounds that showed versatile pharmacological activities including anti-inflammatory [10,11,12], antioxidant [12] and anti-carcinogenic [10,13].

On the other hand, another natural scaffold was chosen for the design of the other fragment of our hybrid molecules. Vanillin (4-hydroxy-3-methoxybenzaldehyde) and isovanillin (3-hydroxy-4-methoxybenzaldehyde) are aromatic aldehydes extensively studied for their potential use as anti-cancer agents [14,15]. Moreover, their nucleus has also been functionalized to create derivatives that maintain or increase this activity [16,17].

The acyl hydrazide–hydrazone moiety served as linker in our hybrid molecules. Hydrazones (–CO–NH–N=CH–) typically result from the condensation of hydrazines with aldehydes or ketones [18] and allow for facile synthetic modification at carbon or nitrogen centers useful to further modulate their reactivity and physicochemical properties, making them valuable tools in medicinal chemistry. The bioactivity of acyl hydrazones can be attributed to their capacity to establish hydrogen bonds with molecular targets, a property conferred by their nature as Schiff bases. Moreover, several studies have investigated the possible use of coumarins with hydrazide moiety at C-3, highlighting hydrazide–hydrazone (–CO–NH–N=CH–) moiety as a significant structural feature in compounds active as antitumor agents [19,20].

Given these premises, we synthesized 17 new compounds (Figure 1) consisting of the coumarin fragment (3-carboxycoumarin unsubstituted or possibly 8-substituted with OH or OCH_3_ group) linked to vanillin or isovanillin or their phenoxy or benzyloxy derivatives via an acylhydrazone linker. To assess the biological potential of these new compounds, we evaluated their effects in several NB cell line (HTLA-230, IMR32 and SK-N-AS) and in a multidrug-resistant (MDR) cell line (ER-HTLA-230) developed in our laboratories and selected by chronically treating HTLA-230 with etoposide [21,22,23].

## 2. Materials and Methods

### 2.1. Chemistry

All chemicals (reagent grade) were purchased from Merck (formerly Aldrich Chemical). Solvents were reagent-grade. All commercial reagents were used without further purification. Reaction progress was monitored using analytical thin layer chromatography (TLC) on precoated silica gel plates (Merck DC-Alufolien Kieselgel 60; F254, Darmastad, Germany) and the spots were detected under UV light (254 nm). Column chromatography was performed on silica gel (60; 0.063–0.200 mm). Melting point was measured on a Stuart SMP3 micromelting point apparatus and uncorrected. Elemental analyses were performed on a Flash 2000 CHNS (Thermo Scientific, Waltham, MA, USA). ^1^H NMR spectra (400 MHz) and ^13^C spectra (101 MHz) were measured on a JEOL JNM ECZ-400S/L1 spectrometer (Tokyo, Japan) at 25 °C or at variable temperature using DMSO-d6 or CDCl_3_ as solvents. Chemical shifts are reported in ppm (δ) relative to the internal reference tetramethylsilane (TMS), and the splitting patterns are described as follows: s(singlet), d(doublet), dd(doublet of doublets), t(triplet), q(quartet) and m(multiplet). The purity of all final compounds was confirmed to be higher than 95% through ^1^H NMR spectra.

### 2.2. Synthesis

#### 2.2.1. Synthesis of Ethyl 3-Hydrazineyl-3-oxopropanoate (A)

A solution of diethyl malonate (8 g, 50 mmol, 7.6 mL) and hydrazine monohydrate (1 mL, 20 mmol) in absolute ethanol (10 mL) is stirred at room temperature for 12 h. The reaction lead to the formation of both malonohydrazide and malonomonohydrazide as described by Abdel-Aziz et al. [24]. The white precipitate, formed at room temperature, (malonohydrazide, m.p.: 148–150 °C) is removed by filtration and the solution is evaporated under vacuum and cooled at −20 °C overnight to afford the crystallization of a white solid (malonomonohydrazide, **A**) which is separated by filtration and washed with diethyl ether. The remaining solution is reacted again with hydrazine monohydrate (0.5 mL, 10 mmol) in absolute ethanol (10 mL). The entire process is repeated up to three times. The monohydrazide fractions obtained are reunited and recrystallized from ethanol.

White solid. Yield: 34% (calculated on diethyl malonate). M.p.: 69–71 °C. Molecular formula: C_5_H_10_N_2_O_3_. M.W.: 146.15 g/mol. Elemental analysis calcd for C_5_H_10_N_2_O_3_. C% 41.09, H% 6.90, N% 19.17; found C% 40.85, H% 6.94, N% 19.35.

#### 2.2.2. General Procedure for the Synthesis of Phenoxy Aldehydes B1-2

Phenoxy aldehydes were synthetized following the method described by Crowder et al. [25].

Briefly, an equimolar amount of solid NaOH (0.20 g, 5 mmol) and of diphenyliodonium chloride (1.58 g, 5 mmol) are added to vanillin or isovanillin (0.76 g, 5 mmol) in H_2_O (35 mL). The reaction mixture is heated to 100 °C for 24 h. After extraction with diethyl ether (3 × 30 mL), the organic phase is washed with 1N NaOH (3 × 20 mL), dried on Na_2_SO_4_, filtered and evaporated under vacuum. Finally, the aldehydes are purified by chromatography 1:15 SiO_2_/n-Exane/Ethyl Acetate (80:20).


**3-methoxy-4-phenoxybenzaldehyde(B1).**


White solid. M.p.: 38–40 °C. (40–42 °C, [26]). Yield: 68%. Molecular formula: C_14_H_12_O_3_. M.W.: 228.25 g/mol. Elemental analysis calcd for C_14_H_12_O_3_. C% 73.67, H% 5.30; found C% 73.45, H% 5.12.


**4-methoxy-3-phenoxybenzaldehyde (B2).**


White solid. M.p.: 43–45 °C. (48 °C, [27]). Yield: 78%. Molecular formula: C_14_H_12_O_3_. M.W.: 228.25 g/mol. Elemental analysis calcd for C_14_H_12_O_3_. C% 73.67, H% 5.30; found C% 73.82, H% 5.47.

#### 2.2.3. General Procedure for the Synthesis of Benzyloxy and 4-F Benzyloxy Aldehydes B3-6

Benzyloxy and 4-fluorobenzyloxy aldehydes were synthetized following the method described by Ashton et al. [28].

To a solution of vanillin or isovanillin (0.76 g, 5 mmol) in an. DMF (5 mL) were added an. K_2_CO_3_ (0.69 g, 5 mmol) and a catalytic amount of KI (50 mg). After stirring at 60 °C for 20 min, a solution of the appropriate benzyl chloride (5 mmol) in an. DMF (2 mL) was added dropwise. The reaction mixture was heated at 70 °C for 22 h. Then, water (30 mL) was added and extractions with ethyl acetate (2 × 20 mL, 1 × 15 mL) were performed. The organic phases were reunited and washed with 1N NaOH (10 mL) and water (15 mL). The aqueous phases were reunited and re-extracted with ethyl acetate (10 mL). Then, the organic phases were reunited and washed with water (10 mL) and Brine (15 mL), dried on Na_2_SO_4_, filtered and evaporated under vacuum to obtain colorless oils that crystallize at room temperature possibly by adding a few drops of anhydrous diethyl ether.


**4-(benzyloxy)-3-methoxybenzaldehyde(B3).**


White solid. Yield: 91%. M.p.: 61–62 °C. (55–57 °C, [29]). Molecular formula: C_15_H_14_O_3_. M.W.: 242.27 g/mol. Elemental analysis calcd for C_15_H_14_O_3_ C% 74.36, H% 5.82; found C% 74.54, H% 5.86.


*
**3-(benzyloxy)-4-methoxybenzaldehyde**
*
**(B4).**


White solid. Yield: 90%. M.p.: 61–63 °C. (61–63 °C, [30]). Molecular formula: C_15_H_14_O_3_. M.W.: 242.27 g/mol. Elemental analysis calcd for C_15_H_14_O_3_: C% 74.36, H% 5.82; found C% 74.12, H% 5.78.

***4-((4-fluorobenzyl)oxy)-3-methoxybenzaldehyde*** **(B5).**

White solid. Yield: 85%. M.p.: 62–63 °C. (60–62 °C, [29]). Molecular formula: C_15_H_13_FO_3_. M.W.: 260.26 g/mol. Elemental analysis calcd for C_15_H_13_FO_3_: C% 69.22; H% 5.03; found C% 69.04, H% 5.01.


*
**3-((4-fluorobenzyl)oxy)-4-methoxybenzaldehyde**
*
**(B6).**


White solid. Yield: 91%. M.p.: 80–81 °C. Molecular formula: C_15_H_13_FO_3_. M.W.: 260.26 g/mol. Elemental analysis calcd for C_15_H_13_FO_3_: C% 69.23; H% 5.03; found C% 69.38, H% 5.07.

#### 2.2.4. General Procedure for the Synthesis of Intermediate Hydrazides C1–7

A solution of **A** and of the desired aldehyde **B1–6** (or 3,4-dihydroxybenzaldehyde for the synthesis of intermediate **C7**) in EtOH (8 mL) was refluxed for 5 h. All intermediates **C1–7** crystallize in the reaction mixture. The volume was reduced under vacuum and cooled overnight at +4 °C to allow for complete crystallization. Intermediates **C1–7** were then filtered out and washed with cold EtOH.


*
**Ethyl 3-(2-(3-methoxy-4-phenoxybenzylidene)hydrazineyl)-3-oxopropanoate**
*
**(C1).**


White solid. M.p.: 105–108 °C. Yield: 79%. Molecular formula: C_19_H_20_N_2_O_5_. M.W.: 356.38 g/mol. Elemental analysis calcd for C_19_H_20_N_2_O_5_: C% 64.04, H% 5.66, N% 7.86; found C% 63.78, H% 5.63, N% 8.01. ^1^H NMR (400 MHz, DMSO-d6) δ 11.55 (s, 1H, NH), 7.91 (s, 1H, CH=N), 7.64–6.74 (m, 8H, arom), 4.05 (q, J = 7.1 Hz, 2H, OCH_2_), 3.76 (d, J = 6.6 Hz, 3H, OCH_3_), 3.61 (s, 2H, COCH_2_CO), 1.11 (t, J = 7.1 Hz, 3H, CH_2_CH_3_). ^13^C NMR (101 MHz, CDCl_3_) δ 168.45, 157.77, 151.88, 145.95, 143.01, 131.82, 130.37, 123.21, 121.57, 121.06, 117.19, 111.03, 60.97, 56.20, 41.84, 14.57.

***Ethyl 3-(2-(4-(benzyloxy)-3-methoxybenzylidene)hydrazineyl)-3-oxopropanoate*** **(C2).**

White solid. M.p.: 139–41 °C. Yield: 94%. Molecular formula: C_20_H_22_N_2_O_5_. M.W.: 370.41 g/mol. Elemental analysis calcd for C_20_H_22_N_2_O_5_: C% 64.85, H% 5.99, N% 7.56; found C% 64.62, H% 5.95, N% 7.34. ^1^H NMR (400 MHz, CDCl_3_) δ 9.97 (s, 1H. NH), 7.72 (s, 1H, CH=N), 7.46–7.24 (m, 6H, Ar), 7.01 (dt, J = 8.3, 2.5 Hz, 1H, Ar), 6.85 (dd, J = 8.3, 3.2 Hz, 1H, Ar), 5.17 (s, 2H, CH_2_Ar), 4.18 (q, J = 7.1 Hz, 2H, OCH_2_), 3.92 (s, 3H, OCH_3_), 3.74 (s, 2H, CH_2_), 1.22 (t, J = 7.1 Hz, 3H, CH_2_CH_3_). ^13^C NMR (101 MHz, CDCl_3_) δ 169.07, 167.59, 150.39, 150.05, 144.50, 136.66, 128.73, 128.72, 128.12, 128.10, 127.33, 126.96, 121.89, 113.23, 108.85, 70.96, 61.40, 56.11, 41.59, 14.26.


***Ethyl 3-(2-(4-((4-fluorobenzyl)oxy)-3-methoxybenzylidene)hydrazineyl)-3-oxopropanoate** *
**(C3).**


White solid. M.p.: 148–149 °C. Yield: 96%. Molecular formula: C_20_H_21_FN_2_O_5_. M.W.: 388.40 g/mol. Elemental analysis calcd for C_20_H_21_FN_2_O_5_: C% 61.85, H% 5.45, N% 7.21; found C% 61.63, H% 5.48, N% 7.01. ^1^H NMR (400 MHz, CDCl_3_) δ 9.95 (s, 1H, NH), 7.72 (s, 1H, CH=N), 7.48–7.39 (m, 2H), 7.28 (d, J = 1.9 Hz, 1H), 7.11–6.99 (m, 3H), 6.84 (dd, J = 8.3, 3.0 Hz, 1H), 5.12 (s, 2H, CH_2_Ar), 4.19 (q, J = 7.1 Hz, 2H, OCH_2_), 3.92 (s, 3H, OCH_3_), 3.74 (s, 2H, CH_2_), 1.22 (t, J = 7.1 Hz, 3H, CH_2_CH_3_). ^13^C NMR (101 MHz, CDCl_3_) δ 169.06, 167.57, 150.18, 150.10, 144.39, 132.41, 132.38, 129.33, 129.25, 127.16, 121.82, 115.74, 115.55, 113.32, 108.88, 70.36, 61.40, 56.09, 41.59, 14.26.


*
**Ethyl 3-(2-(4-methoxy-3-phenoxybenzylidene)hydrazineyl)-3-oxopropanoate**
*
**(C4).**


White solid. M.p.: 129–131 °C. Yield: 68%. Molecular formula: C_19_H_20_N_2_O_5_. M.W.: 356.38 g/mol. Elemental analysis calcd for C_19_H_20_N_2_O_5_: C% 64.04, H% 5.66, N% 7.86; found C% 64.22, H% 5.69, N% 7.65. ^1^H NMR (400 MHz, DMSO-d6) δ 11.42 (s, 1H, NH), 7.84 (s, 1H, CH=N), 7.44–6.76 (m, 8H), 3.93 (q, J = 7.1 Hz, 2H, OCH_2_), 3.74 (s, 3H, OCH_3_), 3.52 (s, 2H, COCH_2_CO), 1.03 (t, J = 7.1 Hz, 3H, CH_2_CH_3_). ^13^C NMR (101 MHz, CDCl_3_) δ 168.25, 158.25, 144.16, 142.83, 130.41, 130.27, 128.02, 126.04, 122.75, 119.46, 117.11, 116.34, 113.90, 60.88, 56.41, 41.70, 14.44.

***Ethyl 3-(2-(3-(benzyloxy)-4-methoxybenzylidene)hydrazineyl)-3-oxopropanoate*** **(C5).**

White solid. M.p.: 110–112 °C. Yield: 94%. Molecular formula: C_20_H_22_N_2_O_5_. M.W.: 370.41 g/mol. Elemental analysis calcd for C_20_H_22_N_2_O_5_: C% 64.85, H% 5.99, N% 7.56; found C% 65.08, H% 6.02, N% 7.79. ^1^H NMR (400 MHz, CDCl_3_) δ 9.98 (s, 1H, NH), 7.69 (s, 1H, CH=N), 7.50–7.44 (m, 2H, Ar), 7.41–7.25 (m, 4H, Ar), 7.08 (dd, J = 8.3, 1.9 Hz, 1H, Ar), 6.86 (d, J = 8.3 Hz, 1H, Ar), 5.18 (s, 2H, CH_2_Ar), 4.18 (q, J = 7.2 Hz, 2H, OCH_2_), 3.90 (s, 3H, OCH_3_), 3.72 (s, 2H, CH_2_), 1.22 (t, J = 7.1 Hz, 3H, CH_2_CH_3_).^13^C NMR (101 MHz, CDCl_3_) δ 169.07, 167.58, 151.87, 148.57, 144.45, 136.88, 128.67, 128.64, 128.08, 127.67, 127.57, 126.51, 122.39, 111.28, 110.95, 71.02, 61.41, 56.11, 41.39, 14.28.

***Ethyl 3-(2-(3-((4-fluorobenzyl)oxy)-4-methoxybenzylidene)hydrazineyl)-3 oxopropanoate*** **(C6).**

White solid. M.p.: 110–111 °C. Yield: 76%. Molecular formula: C_20_H_21_FN_2_O_5_. M.W.: 388.40 g/mol. Elemental analysis calcd for C_20_H_21_FN_2_O_5_: C% 61.85, H% 5.45 N% 7.21; found C% 62.09, H% 5.42, N% 7.38. ^1^H NMR (400 MHz, CDCl_3_) δ 9.98 (s, 1H, NH), 7.69 (t, J = 2.0 Hz, 1H, CH=N), 7.51–7.38 (m, 2H, Ar), 7.30 (d, J = 1.9 Hz, 1H, Ar), 7.18–6.92 (m, 3H), 6.86 (d, J = 8.3 Hz, 1H, Ar), 5.13 (s, 2H, CH_2_Ar), 4.18 (q, J = 7.1 Hz, 2H, OCH_2_), 3.89 (s, 3H, OCH_3_), 3.72 (s, 2H, CH_2_), 1.22 (t, J = 7.1 Hz, 3H, CH_2_CH_3_). ^13^C NMR (101 MHz, CDCl_3_) δ 167.55, 163.86, 161.41, 151.90, 148.40, 132.60, 129.53, 129.45, 126.49, 122.63, 115.69, 115.64, 115.47, 111.28, 110.90, 70.37, 61.42, 56.10, 41.47, 14.26.


*
**Ethyl 3-(2-(3,4-dihydroxybenzylidene)hydrazineyl)-3-oxopropanoate**
*
**(C7).**


White solid. M.p.: 156–157 °C. Yield: 55%. Molecular formula: C_12_H_14_N_2_O_5_. M.W.: 266.26 g/mol. Elemental analysis calcd for C_12_H_14_N_2_O_5_: C% 54.13, H% 5.30, N% 10.52; found C% 54.27, H% 5.69, N% 10.68. ^1^H NMR (400 MHz, DMSO-d6) δ 11.25 (s, 1H, NH), 9.38 (s, 1H, OH), 9.06 (s, 1H, OH), 7.74 (s, 1H, CH=N), 7.04–6.68 (m, 3H, arom), 4.04 (q, J = 7.1 Hz, 2H, OCH_2_), 3.56 (s, 2H, CH_2_), 1.13 (t, J = 7.1 Hz, 3H, CH_2_CH_3_). ^13^C NMR (101 MHz, DMSO-d6) δ 168.25, 148.27, 146.23, 146.16, 144.39, 120.42, 116.03, 113.39, 60.96, 41.41, 14.57.

#### 2.2.5. General Procedure for the Synthesis of Final Hydrazones **1**–**17**

To a solution of proper intermediate **C1–7** (0.8 mmol) and of the appropriate 2-hydroxybenzaldehyde (0.8 mmol) in abs. EtOH (8 mL), piperidine (0.3 mL, 3 mmol) was added. The reaction mixture was stirred for 5 h under reflux. Then, the reaction mixture was concentrated under vacuum and crystallized at –20 °C. The precipitate was filtered and washed with an. Et_2_O to obtain the desired compounds as yellow solids.

For compounds **13**–**17** (OH at position 8 of the coumarin ring) the amount of piperidine was reduced (0.1 mL, 1 mmol) and the purification required and additional step. In detail, the reaction mixture was allowed to cool down and then a mixture of iced water and hydrochloridric acid (10 mL + 5 mL of 1N HCl) was added to afford the complete precipitation of the compounds. After filtration and washes with water, the compounds are purified by crystallization from ethanol.

As will be discussed in detail later (Chemistry subsection of Result and Discussion, Section 3.1), compounds **1**–**17** exist in solution as an equilibrium of two possible rotamers. In the ^1^H NMR spectra, signals for which the corresponding resonance of the alternative rotamer is detectable are annotated as “and.” Conversely, in the aromatic region, where resonances of the minor rotamer cannot always be clearly resolved, they are reported as follows (e.g.,): m, 12H, Ar and Ar_rotamer (Appendix A).


*
**(E)-N’-(3-methoxy-4-phenoxybenzylidene)-2-oxo-2H-chromene-3-carbohydrazide**
*
**(1).**


Yellow solid. M.p.: 256–261 °C. Yield: 85%. Molecular formula: C_24_H_18_N_2_O_5_. M.W.: 414.42 g/mol. Elemental analysis calcd for C_24_H_18_N_2_O_5_: C% 69.56, H% 4.38, N% 6.76; found C% 67.12, H% 4.41, N% 6.34. ^1^H NMR (400 MHz, DMSO-d6) δ 12.03 and 11.74 (s, 1H, NH), 8.89 and 8.29 (s, 1H, Ar and Ar_rotamer), 8.43 and 7.97 (s, 1H, CH=N), 8.07–6.71 (m, 12H, Ar and Ar_rotamer), 3.80 and 3.57 (s, 3H, OCH_3_). ^13^C NMR (101 MHz, DMSO-d6) δ 160.48, 157.59, 154.45, 149.57, 148.36, 134.89, 131.45, 130.86, 130.44, 125.83, 123.40, 122.10, 121.37, 119.89, 118.98, 117.47, 116.81, 111.15, 56.27.


*
**(E)-N’-(4-(benzyloxy)-3-methoxybenzylidene)-2-oxo-2H-chromene-3-carbohydrazide**
*
**(2).**


Yellow solid. M.p.: 213–214 °C. Yield: 86%. Molecular formula: C_25_H_20_N_2_O_5_. M.W.: 428.44 g/mol. Elemental analysis calcd for C_25_H_20_N_2_O_5_: C% 70.09; H%, 4.71; N% 6.54; found C% 69.85, H% 4.69, N% 6.33. ^1^H NMR (400 MHz, DMSO-d6) δ 11.91 and 11.64 (s, 1H, NH), 8.87 and 8.26 (s, 1H, Ar and Ar_rotamer), 8.34 and 7.88 (s, 1H, CH=N), 8.05–6.92 (m, 12H, Ar and Ar_rotamer), 5.11 and 5.03 (s, 2H, CH_2_Ph), 3.80 and 3.55 (s, 3H, OCH_3_). ^13^C NMR (101 MHz, DMSO-d6) δ 160.50, 158.38, 154.41, 150.55, 150.07, 149.84, 148.20, 137.27, 134.83, 130.82, 129.01, 128.53, 128.49, 127.40, 125.81, 122.72, 119.92, 118.98, 116.78, 113.53, 109.28, 70.38, 56.06, 55.52.


*
**(E)-N’-(4-((4-fluorobenzyl)oxy)-3-methoxybenzylidene)-2-oxo-2H-chromene-3-carbohydrazide**
*
**(3).**


Yellow solid. M.p.: 235–256 °C. Yield: 91%. Molecular formula: C_25_H_19_FN_2_O_5_. M.W.: 428,44 g/mol. Elemental analysis calcd for C_25_H_19_FN_2_O_5_: C% 67.26, H% 4.29, N% 6.28; found C% 67.11, H% 4.32, N% 6.24. ^1^H NMR (400 MHz, DMSO-d6): δ 11.89 and 11.63 (s, 1H, NH), 8.87 and 8.25 (s, 1H, CH=N), 8.34 and 7.89 (s, 1H, Ar and Ar_rotamer), 8.04–6.95 (m, 11 H, Ar and Ar_rotamer), 5.10 and 5.02 (s, 2H, CH_2_Ph), 3.80 and 3.55 (s, 3H, OCH_3_). ^13^C NMR (101 MHz, DMSO-d6) δ: 160.48, 158.39, 154.42, 150.45, 150.06, 149.89, 148.18, 134.82, 133.55, 130.81, 130.75, 130.67, 127.54, 125.81, 122.65, 119.95, 118.99, 116.78, 115.93, 115.72, 113.70, 109.43, 69.70, 56.10.


*
**(E)-N’-(4-methoxy-3-phenoxybenzylidene)-2-oxo-2H-chromene-3-carbohydrazide**
*
**(4).**


Yellow solid. M.p.: 243–244 °C. Yield: 92%. Molecular formula: C_24_H_18_N_2_O_5_. M.W.: 414.42 g/mol. Elemental analysis calcd for C_24_H_18_N_2_O_5_: C% 69.56, H% 4.38, N% 6.76; found C% 69.72, H% 4.15, N% 6.58. ^1^H NMR (400 MHz, DMSO-d6) δ 11.87 and 11.64 (s, 1H, NH), 8.85 and 8.19 (s, 1H, Ar), 8.34 and 7.89 (s, 1H, CH=N), 8.03–6.70 (m, 12H, Ar and Ar_rotamer), 3.78 and 3.71 (s, 3H, OCH_3_). ^13^C NMR (101 MHz, DMSO-d6) δ 160.46, 158.43, 157.88, 154.42, 153.67, 149.22, 148.29, 144.81, 134.85, 130.82, 130.45, 130.19, 127.69, 126.14, 125.81, 123.21, 119.82, 119.29, 118.97, 117.56, 117.17, 116.78, 113.95, 56.45.


*
**(E)-N’-(3-(benzyloxy)-4-methoxybenzylidene)-2-oxo-2H-chromene-3-carbohydrazide**
*
**(5).**


Yellow solid. M.p.: 214–215 °C. Yield: 92%. Molecular formula: C_25_H_20_N_2_O_5_. M.W.: 428,44 g/mol. Elemental analysis calcd for C_25_H_20_N_2_O_5_: C% 70.09; H% 4.71; N% 6.54; found C% 70.27, H% 4.68, N% 6.70. ^1^H NMR (400 MHz, DMSO-d6) δ 11.87 and 11.63 (s, 1H, NH), 8.88 and 8.27 (s, 1H, Ar and Ar_rotamer), 8.33 and 7.86 (s, 1H, CH=N), 8.07–6.87 (m, 12H, Ar and Ar_rotamer), 5.10 and 4.86 (s, 2H, CH_2_Ph), 3.79 and 3.72 (s, 3H, OCH_3_). ^13^C NMR (101 MHz, DMSO-d6) δ 160.49, 158.36, 154.41, 151.87, 150.08, 148.61, 148.20, 137.37, 134.83, 130.81, 128.96, 128.89, 128.51, 128.02, 127.09, 125.81, 123.03, 119.92, 118.98, 116.79, 112.35, 110.97, 70.45, 56.21.


*
**(E)-N’-(3-((4-fluorobenzyl)oxy)-4-methoxybenzylidene)-2-oxo-2H-chromene-3-carbohydrazide**
*
**(6).**


Yellow solid. M.p.: 235–236 °C. Yield: 93%. Molecular formula: C_25_H_19_FN_2_O_5_. M.W.: 446.43 g/mol. Elemental analysis calcd for C_25_H_19_FN_2_O_5_: C% 67.26; H% 4.29; N% 6.28; found C%, 67.19, H%, 4.32, N% 6.24. ^1^H NMR (400 MHz, DMSO-d6) δ: 11.86 and 11.62 (s, 1H, NH), 8.87 and 8.24 (s, 1H, Ar and Ar_rotamer), 8.34 and 7.87 (s, 1H, CH=N), 8.02–6.91 (m, 11H, Ar and Ar_rotamer), 5.09 and 4.84 (s, 2H, CH_2_Ph), 3.79 and 3.72 (s, 3H, OCH_3_). ^13^C NMR (101 MHz, DMSO-d6) δ: 161.18, 160.48, 158.37, 154.42, 151.91, 150.05, 148.54, 148.18, 134.82, 133.67, 130.81, 130.74, 130.66, 127.13, 125.81, 123.12, 119.95, 118.99, 116.78, 115.88, 115.67, 112.43, 111.18, 69.81, 56.24.


*
**(E)-8-methoxy-N’-(3-methoxy-4-phenoxybenzylidene)-2-oxo-2H-chromene-3-carbohydrazide**
*
**(7).**


Yellow solid. M.p.: 239–240 °C. Yield: 75%. Molecular formula: C_25_H_20_N_2_O_6_. M.W.: 444.44 g/mol. Elemental analysis calcd for C_25_H_20_N_2_O_6_: C% 67.56, H% 4.54, N% 6.30; found C% 67.32, H% 4.59, N% 6.44. ^1^H NMR (400 MHz, DMSO-d6) δ 12.03 and 11.73 (s, 1H, NH), 8.86 and 8.26 (s, 1H, Ar and Ar_rotamer), 8.43 and 7.96 (s, 1H, CH=N), 7.59–6.76 (m, 11H, Ar and Ar_rotamer), 3.92 and 3.88 (s, 3H, OCH_3_), 3.79 and 3.57 (s, 3H, OCH_3_). ^13^C NMR (101 MHz, DMSO-d6) δ 160.21, 158.53, 157.59, 151.81, 149.61, 148.59, 146.86, 146.65, 143.73, 131.45, 130.43, 130.33, 125.77, 123.39, 122.09, 121.73, 121.37, 119.96, 119.53, 117.47, 117.03, 116.80, 111.15, 56.77, 56.27.


*
**(E)-N’-(4-(benzyloxy)-3-methoxybenzylidene)-8-methoxy-2-oxo-2H-chromene-3-carbohydrazide**
*
**(8).**


Yellow solid. M.p.: 233–234 °C. Yield: 91%. Molecular formula: C_26_H_22_N_2_O_6_. M.W.: 458,47 g/mol. Elemental analysis calcd for C_26_H_22_N_2_O_6_: C% 68.11; H% 4.84; N% 6.11; found C% 68.32, H% 4.80, N% 6.29. ^1^H NMR (400 MHz, DMSO-d6) δ 11.91 and 11.64 (s, 1H, NH), 8.84 and 8.23 (s, 1H, Ar and Ar_rotamer), 8.34 and 7.87 (s, 1H, CH=N), 7.60–6.86 (m, 11H, Ar and Ar_rotamer), 5.11 and 5.03 (s, 2H, CH_2_Ph), 3.91 and 3.90 (s, 3H, OCH_3_), 3.80 and 3.56 (s, 3H, OCH_3_). ^13^C NMR (101 MHz, DMSO-d6) δ 160.23, 158.34, 150.56, 150.12, 149.85, 148.43, 146.85, 143.70, 137.27, 129.01, 128.53, 128.49, 127.40, 125.76, 122.73, 121.70, 120.02, 119.55, 116.76, 113.55, 109.30, 70.38, 56.77, 56.07.


*
**(E)-N’-(4-((4-fluorobenzyl)oxy)-3-methoxybenzylidene)-8-methoxy-2-oxo-2H-chromene-3-carbohydrazide**
*
**(19).**


Yellow solid. M.p.: 254–255 °C. Yield: 92%. Molecular formula: C_26_H_21_FN_2_O_6_. M.W.: 476.46 g/mol. Elemental analysis calcd for C_26_H_21_FN_2_O_6_: C% 65.54, H% 4.44, N% 5.88; found C% 65.12, H% 4.49, N% 5.92. ^1^H NMR (400 MHz, DMSO-d6): ^1^H NMR (400 MHz, DMSO-D6) δ 11.89 and 11.63 (s, 1H, NH), 8.84 and 8.22 (s, 1H, CH=N), 8.34 and 7.88 (s, 1H, ArHand Ar_rotamer), 7.89–6.95 (m, 10 H, Ar and Ar_rotamer) 5.09 and 5.02 (s, 2H, CH_2_Ph), 3.92 and 3.90 (s, 3H, OCH_3_), 3.80 and 3.56 (s, 3H, OCH_3_). ^13^C NMR (101 MHz, DMSO-d6) δ 159.79, 158.39, 150.19, 148.32, 147.03, 146.57, 142.60, 133.66, 130.46, 128.01, 127.43, 125.73, 122.50, 121.79, 120.16, 119.65, 117.11, 115.85, 115.63, 114.40, 110.35, 70.05, 56.96, 56.38.


*
**(E)-8-methoxy-N’-(4-methoxy-3-phenoxybenzylidene)-2-oxo-2H-chromene-3-carbohydrazide**
*
**(10).**


Yellow solid. M.p.: 228–229 °C. Yield: 74%. Molecular formula: C_25_H_20_N_2_O_6_. M.W.: 444.44 g/mol. Elemental analysis calcd for C_25_H_20_N_2_O_6_: C% 67.56, H% 4.54, N% 6.30; found C% 67.72, H% 4.38, N% 6.12. ^1^H NMR (400 MHz, DMSO-d6) δ 11.87 and 11.64 (s, 1H, NH), 8.82 and 8.17 (s, 1H, Ar and Ar_rotamer), 8.35 and 7.89 (s, 1H, CH=N), 7.59–6.67 (m, 11H, Ar and Ar_rotamer), 3.91 and 3.92 (s, 3H, OCH_3_), 3.78 and 3.71 (s, 3H, OCH_3_). ^13^C NMR (101 MHz, DMSO-d6) δ 160.19, 158.37, 157.88, 153.67, 149.26, 148.54, 146.84, 144.79, 143.69, 130.44, 130.11, 127.68, 126.15, 125.75, 123.21, 121.70, 119.88, 119.52, 119.29, 117.47, 117.16, 116.77, 113.93, 56.75, 56.44.


*
**(E)-N’-(3-(benzyloxy)-4-methoxybenzylidene)-8-methoxy-2-oxo-2H-chromene-3-carbohydrazide**
*
**(11).**


Yellow solid. M.p.: 228–229 °C. Yield: 93%. Molecular formula: C_26_H_22_N_2_O_6_. M.W.: 458.47 g/mol. Elemental analysis calcd for C_26_H_22_N_2_O_6_: C% 68.11; H% 4.84; N% 6.11; O% 20.94; found C% 68.11, H% 4.84, N% 6.11. ^1^H NMR (400 MHz, DMSO-d6) δ: 11.88 and 11.63 (s, 1H, NH), 8.84 and 8.24 (s, 1H, Ar and Ar_rotamer), 8.34 and 7.86 (s, 1H, CH=N), 7.60–6.86 (m, 11H, Ar and Ar_rotamer), 5.10 and 4.83 (s, 2H, CH_2_Ph), 3.91 and 3.75 (s, 3H, OCH_3_), 3.79and 3.72 (s, 3H, OCH_3_). ^13^C NMR (101 MHz, DMSO-d6) δ: 160.22, 158.31, 151.87, 150.12, 148.61, 148.43, 146.85, 143.69, 137.37, 128.96, 128.88, 128.51, 128.17, 127.08, 125.75, 123.03, 121.69, 120.01, 119.55, 116.74, 115.38, 112.35, 110.97, 70.45, 56.76, 56.21.


*
**(E)-N’-(3-((4-fluorobenzyl)oxy)-4-methoxybenzylidene)-8-methoxy-2-oxo-2H-chromene-3-carbohydrazide**
*
**(12).**


Yellow solid. M.p.: 219–220 °C. Yield: 93%. Molecular formula: C_26_H_21_FN_2_O_6_. M.W.: 476.46 g/mol. Elemental analysis calcd for C_26_H_21_FN_2_O_6_: C% 65.54, H% 4.44, N% 5.88; found C% 65.18, H% 4.42, N% 5.84. ^1^H NMR (400 MHz, DMSO-d6) δ: 11.86 and 11.62 (s, 1H, NH), 8.84 and 8.23 (s, 1H, CH=N), 8.34 and 7.86 (s, 1H, Ar and Ar_rotamer), 7.90–6.88 (m, 10H, Ar and Ar_rotamer), 5.09 and 4.82 (s, 2H, CH_2_Ph), 3.92 and 3.76 (s, 3H, OCH_3_), 3.79 and 3.71 (s, 3H, OCH_3_). ^13^C NMR (101 MHz, DMSO-d6) δ: 160.21, 158.33, 151.92, 150.10, 148.53, 148.42, 146.87, 143.72, 130.75, 130.66, 127.12, 125.76, 123.13, 121.71, 120.04, 119.56, 116.79, 115.88, 115.66, 112.42, 111.18, 69.81, 56.79, 56.24.


*
**(E)-8-hydroxy-N’-(3-methoxy-4-phenoxybenzylidene)-2-oxo-2H-chromene-3-carbohydrazide**
*
**(13).**


Yellow solid. M.p.: 243–246 °C. Yield: 93%. Molecular formula: C_24_H_18_N_2_O_6_. M.W.: 430.42 g/mol. Elemental analysis calcd for C_24_H_18_N_2_O_6_: C% 66.97, H% 4.22, N% 6.51; found C% 66.78, H% 4.18, N% 6.33. ^1^H NMR (400 MHz, DMSO-d6) δ 12.01 and 11.75 (s, 1H, NH), 10.47 and 10.30 (s, 1H, OH), 8.83 and 8.21 (s, 1H, Ar and Ar_rotamer), 8.43 and 7.96 (s, 1H, CH=N), 7.56–6.76 (m, 11H, Ar and Ar_rotamer), 3.80 and 3.58 (s, 3H, OCH_3_). ^13^C NMR (101 MHz, DMSO-d6) δ 160.42, 158.65, 157.59, 151.81, 149.52, 148.83, 146.63, 145.07, 143.08, 131.47, 130.43, 130.33, 125.81, 123.38, 122.07, 121.37, 120.90, 120.66, 119.93, 119.59, 117.46, 117.03, 111.16, 56.27.


*
**(E)-8-hydroxy-N’-(4-methoxy-3-phenoxybenzylidene)-2-oxo-2H-chromene-3-carbohydrazide**
*
**(14).**


Yellow solid. M.p.: 270–271 °C. Yield: 62%. Molecular formula: C_24_H_18_N_2_O_6_. M.W.: 430.42 g/mol. Elemental analysis calcd for C_24_H_18_N_2_O_6_: C% 66.97, H% 4.22, N% 6.51; found C% 66.85, H% 4.19, N% 6.30. ^1^H NMR (400 MHz, DMSO-d6) δ 11.85 and 11.66 (s, 1H, NH), 10.46 and 10.34 (s, 1H, OH), 8.78 and 8.12 (s, 1H, Ar and Ar_rotamer), 8.35 and 7.89 (s, 1H, CH=N), 7.60–6.73 (m, 11H, Ar and Ar_rotamer), 3.78 and 3.71 (s, 3H, CH_3_). ^13^C NMR (101 MHz, DMSO-d6) δ 160.40, 158.51, 157.87, 153.65, 149.16, 148.76, 145.05, 144.80, 143.04, 130.45, 127.70, 126.13, 125.78, 123.21, 120.87, 120.63, 119.92, 119.55, 119.27, 117.17, 113.94, 56.45.


*
**(E)-N’-(3-(benzyloxy)-4-methoxybenzylidene)-8-hydroxy-2-oxo-2H-chromene-3-carbohydrazide**
*
**(15).**


Yellow solid. M.p.: 258–260 °C. Yield: 75%. Molecular formula: C_25_H_20_N_2_O_6_. M.W.: 444.44 g/mol. Elemental analysis calcd for C_25_H_20_N_2_O_6_: C% 67.56, H% 4.54, N% 6.30; found C% 67.34, H% 4.57, N% 6.12. ^1^H NMR (400 MHz, DMSO-d6) δ: 11.85 and 11.65 (s, 1H, NH), 10.46 and 10.35 (s, 1H, OH), 8.80 and 8.19 (s, 1H, Ar and Ar_rotamer), 8.33 and 7.86 (s, 1H, CH=N), 7.51–6.88 (m, 11H, Ar and Ar_rotamer), 5.10 and 4.88 (s, 2H, CH_2_Ph), 3.79 and 3.73 (s, 3H, OCH_3_). ^13^C NMR (101 MHz, DMSO-d6) δ: 160.43, 158.44, 151.86, 150.03, 148.66, 148.61, 145.06, 143.05, 137.38, 128.96, 128.87, 128.51, 127.89, 127.11, 125.79, 123.01, 120.85, 120.62, 119.94, 119.66, 112.36, 110.98, 70.45, 56.21.


*
**(E)-N’-(3-((4-fluorobenzyl)oxy)-4-methoxybenzylidene)-8-hydroxy-2-oxo-2H-chromene-3-carbohydrazide**
*
**(16).**


Yellow solid. M.p.: 282–284 °C. Yield: 74%. Molecular formula: C_25_H_19_FN_2_O_6_. M.W.: 462.43 g/mol. Elemental analysis calcd for C_25_H_19_FN_2_O_6_: C% 64.93, H% 4.14, N% 6.06; found C% 65.12, H% 4.17, N% 6.28. ^1^H NMR (400 MHz, DMSO-d6) δ 11.84 and 11.64 (s, 1H, NH); 10.44 (s, 1H, OH), 8.80 and 8.19 (s, 1H, Ar and Ar_rotamer), 8.34 and 7.87 (s, 1H, CH=N), 7.64–6.84 (m, 9H, Ar and Ar_rotamer), 5.09 and 4.86 (s, 3H, OCH_3_), 3.79 and 3.72 (s, 3H, OCH_3_). ^13^C NMR (101 MHz, DMSO-d6) δ NMR (101 MHz, DMSO-d6) δ 160.42, 158.45, 151.90, 150.00, 148.66, 148.53, 145.02, 143.05, 133.66, 130.75, 130.67, 127.14, 125.79, 123.11, 120.83, 120.65, 119.94, 119.64, 115.88, 115.67, 112.42, 111.17, 69.80, 56.23.


*
**(E)-N’-(3,4-dihydroxybenzylidene)-8-hydroxy-2-oxo-2H-chromene-3-carbohydrazide**
*
**(17).**


Yellow solid. M.p.: >280 °C. Yield: 57%. C_17_H_12_N_2_O_6_. M.W.: 340.29 g/mol. Elemental analysis calcd for C17H12N2O6: C% 60.00, H% 3.55, N% 8.23; found C% 60.00, H% 3.55, N% 8.23. ^1^H NMR (400 MHz, DMSO-d6) δ 11.66 and 11.54 (s, 1H, NH), 10.44 and 10.31 (s, 1H, OH), 9.40 and 9.30 (s, 1H, OH), 9.26 and 9.10 (s, 1H, OH), 8.80 and 8.16 (s, 1H, Ar and Ar_rotamer), 8.21 and 7.81 (s, 1H, CH=N), 7.38 (dd, J = 6.8, 2.5 Hz, 1H, Ar and Ar_rotamer), 7.26–7.10 (m, 3H, Ar and Ar_rotamer), 6.95 (dd, J = 8.2, 2.0 Hz, 1H, Ar), 6.75 (d, J = 8.1 Hz, 1H, Ar). ^13^C NMR (101 MHz, DMSO-d6) δ 160.43, 158.31, 150.30, 148.88, 148.60, 146.25, 145.05, 143.04, 125.92, 125.77, 121.55, 120.81, 120.61, 119.96, 119.66, 116.12, 113.51.

### 2.3. Chemical Radical Scavenging Activity

The antioxidant activity of compounds **1**–**17** was measured using DPPH assay (Sigma-Aldrich, Milan, Italy) [31,32]. A total of 0.1 mL of was mixed with 3.9 mL of *2,2-diphenyl-1-picrylhydrazyl* (DPPH) methanol solution (65 μM). Absorbance was measured at 517 nm after reacting for 30 min in the dark. The linear calibration curve was obtained using Trolox standards (ranging between 20 and 200 mg/L, R^2^ = 0.9955). The results were calculated as Trolox equivalent in mg/L, and the percentage of antioxidant activity (AA%) was calculated from the ratio of decreasing absorbance of sample solution (*A*0–*As*) to absorbance of blank DPPH solution (A), as expressed in Equation (1) [33].
$$AA\%=\frac{A0-As}{A0}\times 100$$

All analyses were performed in triplicate (n = 3), and values are given ± standard deviation (SD).

### 2.4. Biological Methods

Etoposide was purchased from Calbiochem (Merk KGaA, Darmstadt, Germany), Ferrostatin-1, Liprostatin-1, RSL3 and erastin were from Sigma-Aldrich (St. Louis, MO, USA). Stock solutions of these compounds were prepared using sterile water or dimethyl sulfoxide (DMSO, Sigma-Aldrich, St. Louis, MO, USA) as solvent.

#### 2.4.1. Cell Cultures

All experiments were performed using malignant NB cells IMR32 and HTLA-230 with MYCN amplification were obtained from G. Gaslini Institute, Genoa, Italy, the etoposide-resistant cell line (ER) was selected and characterized as previously described [21,22], SK-N-AS without MYCN amplification were obtained from Biologic Bank and Cell Factory, IRCCS Policlinico San Martino, Genoa, Italy. HaCat, a human keratinocytes cell line (kindly provided by Dr. Marco Ponassi, Biologic Bank and Cell Factory, IRCCS Policlinico San Martino, Genoa, Italy) and 3T3, mouse embryonic fibroblast (Sigma-Aldrich) were used as non-malignant cells. Cells were periodically tested for mycoplasma contamination (Mycoplasma Reagent Set, Aurogene s.p.a, Pavia, Italy). Cells were cultured in RPMI 1640 (Euroclone SpA, Pavia, Italy) and supplemented with 10% fetal bovine serum (FBS; Euroclone SpA, Pavia, Italy), 2 mM glutamine (Euroclone SpA, Pavia, Italy), 1% penicillin/streptomycin (Euroclone SpA, Pavia, Italy), 1% sodium pyruvate (Sigma-Aldrich, Saint Louis, MO, USA) and 1% amino acid solution (Sigma-Aldrich, Saint Louis, MO, USA).

#### 2.4.2. Cell Treatments

HTLA-230, ER and HaCat were treated for 72 h with 20 µM **1**–**17**. Moreover, in another series of experiments, HTLA-230 and ER cells were treated with ferrostatin and liprostatin, as ferroptosis inhibitors, and RSL3 and erastin, as ferroptosis inducers, respectively at doses of 2.5 µM, 1 µM, 50 nM and 2.5 µM for 72 h, alone or in combination with 20 µM **5**, **9**, **12** and **17**. Furthermore, the analysis was extended by treating all malignant and non- malignant cell lines with increasing concentrations of compounds **5**, **9** and **12** for 72 h. In order to exclude the interference of DMSO, employed to solubilize these compounds, cells were exposed to the highest concentration used and the pilot studies were carried out in parallel with all performed analyses.

#### 2.4.3. Cell Viability Assay (MTS)

In order to assess cell viability/proliferation, the *3-(4,5-dimethylthiazol-2-yl)-5-(3-carboxymethoxyphenyl)-2-(4-sulfophenyl)-2H-tetrazolium* (MTS) CellTiter 96^®^ AQueous One Solution cell proliferation assay (Promega, Madison, WI, USA) was used according to the manufacturer’s instructions. Briefly, cells (10,000 cells/well) were seeded and treated for 72 h in 96-well plates (Corning Incorporated, Corning, New York, NY, USA) as above reported. Next, cells were incubated for 1h with 20 µL of CellTiter in standard conditions (37 °C in humidified incubator with 5% CO_2_). Absorbance was recorded at 490 nm using a microplate reader (EL-808, BIO-TEK Instruments Inc., Winooski, VT, USA).

#### 2.4.4. Total Protein Extraction and Quantification

Cells were detached in lysis buffer (50 mM Tris HCl, 150 mM NaCl, 46 2 mM EDTA, 1 mM EGTA, 50 mM EGF, 1 mM PMSF) containing a mixture of 7× protease inhibitors (complete Mini Protease, Roche) and 1% Triton X-100 (Sigma). The suspension was further lysed by passing it 10 times through a 25 Gauge needle equipped syringe and centrifuged at 15.000 g for 10 min. at 8 °C (Heraeus Centrifuge Biofuge 28RS). The supernatant was collected and its protein concentration was determined by PierceTM BCA Protein Assay Kit (ThermoFisher Scientific, Waltham, MA, USA). Total protein amounts of the samples were calculated in respect to a calibration line obtained by using bovine serum albumin (BSA) at standard concentrations (mg/mL).

#### 2.4.5. Western Blot Analysis

Immunoblots were carried out according to standard methods [34] using rabbit antibody anti-PARP (Cell Signaling Technology Inc., Danvers, MA, USA, Upstate, Lake Placid, NY, USA) and mouse monoclonal antibody anti-β-actin (Sigma-Aldrich). Anti-rabbit (Cell signaling Technologies) and Anti-mouse (Invitrogen, ThermoFisher, Waltham, MA, USA) secondary antibodies coupled with horseradish peroxidase were utilized. The signal was detected using ECL chemiluminescence system detection kit (Pierce™ ECL Western Blotting Substrate, ThermoFisher).

#### 2.4.6. ROS Production

ROS production was evaluated by adding *2′-7′ dichlorofluorescein-diacetate* (DCFH-DA; Sigma-Aldrich) at the end of the treatments, as previously reported [21,35]. HTLA-230, ER and HaCat cells (10,000/well) were seeded and treated for 72 h in 96-well plates (Corning) as above described. After treatments, cells were stained with DCFH-DA for 30 min at 37 °C. Then, cells were incubated with 90% DMSO for 10 min in the dark. The generated fluorescence intensity was monitored with Perkin Elmer fluorometer (Perkin Elmer Life and Analytical Sciences, Shelton, CT, USA) at 485/530 nm excitation/emission and was proportional to the ROS amount. Values were normalized to the protein content.

#### 2.4.7. Lipid Peroxidation Assay

Lipid peroxidation was measured using the Image-iT™ Lipid Peroxidation Kit (Thermo Fisher), which allows for detection of lipid peroxidation in live cells by oxidation of BODIPY™ 581/591 C11 reagent. HTLA-230, ER and HaCat cells (10,000/well) were seeded and treated for 72 h in 96-well plates (Corning) as above described. Afterwards, BODIPY reagent was added and incubated for 30 min at 37 °C. Fluorescence was monitored with Perkin Elmer fluorometer (Perkin Elmer Life and Analytical Sciences, Shelton, CT, USA) first at 580/590 nm excitation/emission and then at 485/520 nm, and the ratio was used for data analysis.

#### 2.4.8. Senescent Cell Evaluation

Senescent cells were detected using CellEvent Senescence Green Detection Kit (Invitrogen, ThermoFisher) following the manufacturer’s instructions. Briefly, HTLA and ER cells (10,000/well) were seeded and treated for 72 h in 96-well plates (Corning) as above reported. Then, cells were incubated with fixation solution for 10 min at room temperature and protected from light. After washing with 1% BSA in PBS, the working solution was added and incubated for 2 h at 37 °C without CO_2_ and protected from light. After incubation, the working solution was removed, and fluorescence was monitored with Perkin Elmer fluorometer (Perkin Elmer Life and Analytical Sciences, Shelton, CT, USA) at 485/530 nm excitation/emission.

#### 2.4.9. Clonogenic Assay

HTLA-230 and ER cells (150/well) were seeded and treated for 72 h in six-well plates (Corning). Subsequently, the medium was changed and the cells were maintained in drug-free medium for 20 days. Cells were then fixed with methanol and stained with crystal violet (0.5% in water with 50% methanol). Colonies containing more than 50 cells were visualized by a Leica DMIRB microscope (Leica, Wetzlar, Germany) and the images were acquired with a Nikon Coolpix L22 camera (Nikon Corporation, Tokyo, Japan).

#### 2.4.10. Statistical Analysis

The results were expressed as mean ± SEM of at least four independent experiments. The statistical significance of the parametric differences between the experimental data sets was assessed by one-way ANOVA and Dunnett’s test for multiple comparisons. *p* < 0.05 was considered statistically significant.

## 3. Results and Discussion

### 3.1. Chemistry

For the synthesis of the new compounds, it was first necessary to prepare compounds **A** (hydrazonic fragment) and **B** (fragment derived from vanillin or isovanillin), which served as starting reagents for the synthesis of hydrazide intermediates **C**. The synthetic procedures for obtaining intermediates **A** and **B** are reported in Figure 1. The malonic ester was reacted with hydrazine monohydrate in ethanol, leading to the formation of both malonohydrazide and malonomonohydrazide, as described by Abdel-Aziz et al. [24] and separated through fractional crystallization. Malonohydrazide was discarded, while malonomonohydrazide was subsequently reacted with aldehydes **B**.

Phenoxy, benzyloxy, and 4-fluoro-benzyloxy aldehydes (**B**) were prepared from vanillin or isovanillin following the procedures outlined in Figure 1 and as previously described in the literature [25,28].

Intermediate **A** was then reacted with aldehydes **B** in absolute ethanol in the presence of piperidine to obtain the linear acylhydrazones **C1–7** (Figure 2), which crystallized directly from the reaction mixture. The final compounds **1**–**17** were synthesized by reacting intermediates **C1**–**7** with the appropriate 2-hydroxybenzaldehyde (Figure 2) in ethanol with the addition of piperidine. The products crystallized directly from the reaction mixture and were obtained as yellow crystals in good yields (>70%). Compounds **13**–**17** (R = OH) were also obtained as yellow crystals, but under modified conditions: a lower amount of piperidine was used, followed by acidic treatment. This additional step explains the reduced yields (30–50%). The intermediates **C1–7** and all final compounds **1**–**17** were characterized by TLC, melting point, elemental analysis and NMR spectra (^1^H and ^13^C).

Compound **17** differs from the others since it lacks phenyl, benzyl, or 4-fluorobenzyl substituents and is instead characterized by the presence of three hydroxyl groups (R = R_1_ = R_2_ = OH). This compound was synthesized as a reference to highlight the importance of an additional aromatic ring. Moreover, since free radical scavenging activity (as assessed by the DPPH assay) is strongly influenced by the presence of free hydroxyl groups—and this compound bears such groups in all positions—it serves as a structural analogue reference with marked antioxidant activity.

In the ^1^H-NMR spectra of all target compounds (**1**–**17**), the N-H bond signal was described at a chemical shift that ranges from d 11.75–11.62, characteristic of the E isomer. The absence of a signal above δ 14.00 demonstrates that the Z isomer is not formed. The N=CH signal was also described at a range of chemical shift of 8.43–8.22, confirming that the E isomer is the only single product formed by the reaction.

Moreover, it is interesting to underline the presence of duplicate signals caused by the possible geometric stereoisomerism in the N-acylhydrazone linker. For example, next to the signals of the prevalent rotamer (**15** ap, Figure 2b) for N-H bond (11.75-11.62), N=CH (8.43-8.22), aromatic H-4 of coumarin nucleus (8.89-8.77), corresponding signals are present at chemical shift values (δ) 12.03-11.64 (N-H), 7.96-7.81 (C=CH), 8.29-8.11 (H-4). The total of the integrals of each pair of signals corresponds to one hydrogen. This chemical behavior has already been studied by several authors and described in the literature [36,37,38]. To confirm that compounds **1**–**17** exist as a mixture of rotamers at room temperature, sequential ^1^H NMR spectra of compound **15**, a compound taken as an example, were obtained at different increasing temperatures (25, 35, 50, 60, and 70 °C). As Figure 2a shows, the signals tend to become closer and coalesce as the temperature increases, as expected in the presence of the equilibrium between rotamers.

### 3.2. Radical Scavenging Activity

Table 1 reports the radical scavenging activity values obtained for the three most active compounds in the biological assays evaluated trough DPPH assay. Vanillin and 3-cumarinic acid are reported as references. All analyses were performed in triplicate. The results were reproducible although no statistically significant differences were observed among the compounds tested (ANOVA, *p* > 0.05).

Compounds **5**, **9**, and **12** retain some direct radical scavenging activity, while compound **17** alone, owing to its catechol group, appears to possess superior radical scavenging activity. This activity is directly linked to the compounds’ chemical structure and may not be related to their activity within the cellular environment, where the redox state is regulated by complex biochemical pathways.

### 3.3. Biological Results

#### 3.3.1. Compounds **5**, **9** and **12** Are Able to Reduce the Growth of Cancer Cells but Not That of Healthy Keratinocytes (HaCat)

Both NB cell lines (HTLA-230 and ER) and healthy keratinocytes (HaCat) were exposed to all tested compounds (20 µM) for 72 h. As shown in Figure 3, nine compounds demonstrated a reduction in cancer cell growth, and in particular were more effective on ER cells rather than on HTLA-230. In fact, among the compounds tested, **16** was the most effective in reducing the growth of ER and HTLA cells by 30% and 20%, respectively, and that of HaCat cells by 20%. A similar effect was observed in cells treated with **1,** while **15** was cytotoxic for ER and HaCat cells (Figure 3). Compounds **7** and **2** were ineffective on HaCat and exerted a limited effect on both NB cell lines. Interestingly, **5**, **9** and **12** were able to reduce the growth of HTLA and ER cells by 10–15% and 20–25%, respectively, without affecting the viability of HaCat cells. Furthermore, **17** was found on one hand to reduce the growth of ER cells by 18% and on the other, to stimulate the cell proliferation of HTLA and HaCat (Figure 3).

#### 3.3.2. Compounds **5**, **9** and **12** Stimulate ROS Overproduction in NB Cells but Not in Healthy Cells

Since the resistance to etoposide and other chemotherapeutics of ER cells was found to be dependent on the increase in their antioxidant response to the prooxidant effect of the drugs [21], the evaluation of the redox state was considered useful to investigate the mechanisms of action of **1**–**17** compounds. As shown in Figure 4, compound **12** was the most efficient in stimulating ROS production in all three tested cell lines and in particular in ER cells in which it induced a 50% increase in ROS levels in respect to untreated cells. Compound 2 stimulated ROS production only in HaCat cells with a 30% increase in respect to untreated cells, while **1** and **7** only partially increased ROS in both NB cells and **9** only in HTLA (Figure 4). Instead, **5, 9,** and **12** were able to induce ROS over-production in HTLA and ER cells without changes in HaCat cells. In particular, especially **9** and **12** were observed to stimulate ROS production by 30% in HTLA and by 35–40% in ER cells. Furthermore, **17** was not able to increase ROS production in NB cells, leading to a reduction in HaCat (Figure 4).

Based on the results obtained, further analyses were performed on both NB cell lines exposed to **5**, **9**, **12** and **17**.

#### 3.3.3. Compounds **5**, **9**, **12** and **17** Do Not Induce Apoptotic Death

To shed light on the mechanisms underlying the effect of **5**, **9**, **12** and **17** on the survival of cancer cells, the levels of full length (116 kDa) and cleaved (89 kDa) PARP, a known marker of apoptosis [39], were evaluated. As shown in Figure 5, all compounds did not induce PARP cleavage in either HTLA or ER cells.

#### 3.3.4. Compounds **5**, **9**, **12** and **17** Do Not Induce Lipoperoxidation and Compound **17** Slightly Stimulates It Only in ER Cells

To better characterize cell damage, the involvement of membrane lipid oxidative degradation was evaluated by analyzing lipid peroxidation (LPO) [40]. As shown in Figure 6, only **17** was able to increase LPO (+10%) in ER cells, while **5**, **9**, **12** and **17** reduced LPO by 15–20% in HTLA cells, in respect to untreated cells (Ctr).

#### 3.3.5. Compounds **5**, **9**, **12** and **17** Do Not Induce Ferroptotic Death

Based on the evidence that **5**, **9** and **12** induced ROS overproduction and non-apoptotic death, the involvement of ferroptosis, a newly described type of programmed cell death, has been considered. For this purpose, both cell populations were exposed to ferroptosis-inhibitory agents (ferrostatin or liprostatin) and to ferroptosis-inducing compounds (RSL3 or erastin) for 72 h, either given alone or co-administered with **5**, **9**, **12** and **17**. As reported in Figure 7a, ferrostatin or liprostatin, given alone, did not alter the viability of both cell populations, and the co-administration with **5**, **9**, **12** or **17** was not able to prevent the cytotoxic effects of the compounds.

On the other hand, erastin and RSL3, given alone reduced the viability of HTLA cells by 52% and 45%, respectively and of ER cells by 20% and 25%, respectively (Figure 7b), and the co-treatment with the compounds did not further increase the cytotoxic action of compounds **5**, **9**, **12** and **17**. In fact, although a reduction in viability was observed in co-treated HTLA cells, the effect is due to the presence of erastin or RSL3, which directly reduced cell viability, and not to a synergistic action with the tested compounds.

Taken together, these results demonstrate that **5**, **9**, **12** and **17** limited NB growth without inducing ferroptotic death.

#### 3.3.6. Compound **5**, **9**, and **12** Decrease NB Cell Growth by Inducing Senescence

Mounting evidence suggests that cellular senescence can act as a tumor suppressor mechanism able to prevent the growth of NB cells [41,42,43,44,45]. For this reason, the presence of senescent cells was evaluated in HTLA and ER cells treated with **5**, **9**, **12** and **17** for 72 h. As shown in Figure 8, compound **5**, **9**, and **12** induced a 50% increase in the senescent positive cells in HTLA, a 40% (**5** and **9**) and 25% (**12**) increase in ER cells, while exposure to compound **17** did not induce the appearance of senescence in both NB cell populations.

#### 3.3.7. Compound **5** Decreases Cell Growth by Reducing the Cellular Clonogenic Potential

To further investigate the cytotoxic effects of the most active compounds tested, the reproductive activity of both treated cell populations was assessed by clonogenic assay. As shown in Figure 9, only compound **5** was able to markedly reduce colonies formation in both cell populations, while compound **17** exerted a similar effect only in HTLA parental cells. Noteworthy, the other compounds were totally unable to counteract the clonogenic potential of HTLA and ER cells.

#### 3.3.8. Compounds **5**, **9**, and **12** Have Substantial Selectivity for NB Cells Versus Non-Malignant Cells

To further characterize compounds **5**, **9**, and **12**, the viability analysis was carried out also in SK-N-AS, a NB cell line without MYCN amplification, and in IMR32, NB cell line with MYCN amplification. In parallel, similar experiments were conducted on HaCat, a human keratinocyte cell line, and on 3T3, an embryonic fibroblast cell line. All cell lines tested, including HTLA and ER, were treated for 72 h with increasing doses of the three compounds. Subsequently, the IC_50_ value of each compound was calculated for each cell line (Table 2), and these values were then used to evaluate the selectivity index (SI) of the compounds (Table 3).

Compounds **5**, **9**, and **12** demonstrated substantial selectivity for NB cells vs. non-malignant cells. In detail, compound **9** showed the highest selectivity, with SI values reaching ~8–13 in ER and 25–39 in IMR-32 cells, while compound **12** showed pronounced selectivity in ER, SK-N-AS, and IMR-32 cells (SI ≈ 13–14). Compound **5** showed a moderate but consistent selectivity towards all NB cell lines.

### 3.4. ADME Properties Prediction

Predictions of pharmacokinetic properties were made using SwissADME software (www.swissadme.ch accessed on 22 December 2025), and the values are reported in Table 4.

All the compounds do not violate Lipinski’s rules as they have a molecular weight lower than 500, LogP below 4.15, N or O number less than 10, and NH or OH number less than 5. From the point of view of physicochemical properties, all compounds have a number of hydrogen bond acceptors between 6 and 8, and a number of hydrogen bond donors equal to 1 or 2, except for compound **17**, where it is equal to 4 (R = OH the ex-aldehyde nucleus is catechol, not substituted with Ph-Bn or 4-FBn). In addition, the topological polar surface area (TPSA) is lower than 140 Å^2^ in all compounds.

All compounds have a predicted high gastrointestinal absorption and would not be substrates of P-glycoprotein. None of the compounds simultaneously inhibit all the cytochromes considered in the pharmacokinetic parameters and compound **17** does not inhibit any of the considered cytochrome. The bioavailability score is 0.55 for all compounds, which combined with 0 Lipinski rule violations, indicates a good probability of oral bioavailability.

## 4. Conclusions

This work reports the fragment-based design, synthesis, and biological evaluation of seventeen novel coumarin–vanillin/isovanillin hybrids acylhydrazones. By combining two natural scaffolds with intrinsic antioxidant and anticancer properties, our goal was to obtain multifunctional molecules able to modulate oxidative stress and inhibit the growth of aggressive NB cells, including multidrug-resistant sublines.

Antioxidant assays revealed that most compounds did not exhibit strong radical-scavenging activity, largely due to hydroxyl functionalization on the vanillin or isovanillin moiety. This interpretation is supported by the behavior of compound **17**, which retains an unmodified catechol group and showed higher activity. In contrast, the hydroxyl substituent at position 8 of the coumarin ring appeared to have little effect on radical-scavenging capacity.

Cell-based studies highlighted that several of the newly synthesized hybrids exerted significant and selective antiproliferative activity toward MYCN-amplified HTLA-230 NB cells and their multidrug-resistant counterpart (ER cells) while not affecting non-malignant keratinocytes (HaCat). In particular, compounds **5**, **9**, and **12** emerged as the most active derivatives, inducing a 10–25% reduction in NB cell viability and concomitantly triggering a robust increase in intracellular ROS levels, especially in multidrug-resistant cells. Interestingly, these effects were not mediated by apoptotic pathways, lipid peroxidation, or ferroptosis, as demonstrated by the insensitivity of treated cells to ferroptosis modulators. Instead, mechanistic analyses indicated that the antiproliferative effects were associated with the induction of a senescent phenotype and with a marked reduction in clonogenic activity in HTLA and ER cells exposed to compound **5**. Furthermore, the high SI values found in the MDR NB cell line represent a key strength of our study. Indeed, the SI values demonstrate a substantial discrimination between malignant and non-malignant cells, suggesting that these compounds possess an intrinsically favorable safety profile. Therefore, based on these considerations, these compounds may be promising for further therapeutic development aimed at offering new therapeutic options for patients refractory to current therapies. From a structure–activity relationship perspective, the comparative analysis between the active compounds and compound **17**, which lacks the second aromatic ring in the ex-aldehyde portion, underscored the critical contribution of this structural fragment to the antitumor efficacy. These findings provide useful insights into the design of future analogs and highlight the potential of hydrazone linkers to preserve or enhance the bioactivity of parent scaffolds while providing chemical versatility for further optimization. Overall, this data would seem to indicate that these compounds have the potential to lead to the development of compounds characterized by adequate physicochemical properties and good drug-likeness.

In conclusion, this study demonstrates that molecular hybridization proves to be an effective strategy for generating new molecules capable of modulating oxidative stress and targeting additional antitumor pathways. The ability of our new hybrid molecules to selectively impair NB and multi-drug-resistant NB cell growth through senescence while preserving healthy cells makes them attractive for further in-depth biological studies and structural optimization aimed at preclinical development.

## Data Availability

The data presented in this study and different from those reported in the Appendix A are available on request from the corresponding author.

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
