# Peer review of "Design and Synthesis of New Coumarin Hybrids Active Against Drug-Sensitive and Drug-Resistant Neuroblastoma Cells"

_antioxidants, 2025, doi:10.3390/antiox15010031_

Round 1
Reviewer 1 Report
The current manuscript reported a study synthesizing 17 coumarin–(iso)vanillin acyl-hydrazone hybrids and screening them at 20 µM for 72 h against MYCN-amplified HTLA-230 neuroblastoma cells and an etoposide-resistant derivative (ER), plus HaCaT keratinocytes. Compounds 5, 9, 12 show modest, selective growth reduction in NB (≈10–25%), trigger ROS increases, do not cleave PARP, do not behave as ferroptosis inducers under their conditions, and induce senescence; compound 5 also lowers clonogenicity. In general, the chemistry is solidly documented, the biology is preliminary. There are several issues where authors should address before being considered for publication.
- The title choice is reasonable but over-claims “targeting.”
- It is highly recommended that authors provide dose–response curves (≥8-point) for 5/9/12 in HTLA, ER, and at least one additional NB line, where IC₅₀ and selectivity indices vs ≥2 non-malignant, lineage-relevant cell could be computed
- There seems to have some minor typos in the methodology section, consider cleaning them up.
- Reconcile DPPH “antioxidant” outputs with cellular pro-oxidant effects and avoid causal language without bridging experiments.
- The SwissADME summary is fine for a chemistry-forward paper, it should be supportive, not conclusive.
Author Response
Reviewer 1
The current manuscript reported a study synthesizing 17 coumarin–(iso)vanillin acyl-hydrazone hybrids and screening them at 20 µM for 72 h against MYCN-amplified HTLA-230 neuroblastoma cells and an etoposide-resistant derivative (ER), plus HaCaT keratinocytes. Compounds 5, 9, 12 show modest, selective growth reduction in NB (≈10–25%), trigger ROS increases, do not cleave PARP, do not behave as ferroptosis inducers under their conditions, and induce senescence; compound 5 also lowers clonogenicity. In general, the chemistry is solidly documented, the biology is preliminary. There are several issues where authors should address before being considered for publication.
We thank the Reviewer for appreciating the study and, according with the valuable suggestions, we have carried out new experiments to strengthen the biological data.
- The title choice is reasonable but over-claims “targeting.”
In accordance with the Reviewer's comments, the title has been appropriately modified as follows: “Design and synthesis of new coumarin hybrids active against drug-sensitive and drug-resistant neuroblastoma cells”.
- It is highly recommended that authors provide dose–response curves (≥8-point) for 5/9/12 in HTLA, ER, and at least one additional NB line, where IC₅₀ and selectivity indices vs ≥2 non-malignant, lineage-relevant cell could be computed.
We thank the Reviewer for this important suggestion. In accordance with this request, ≥10-point dose-response experiments were performed by treating 4 neuroblastoma cell lines (HTLA, ER, SK-N-AS, and IMR-32) with increasing concentrations of compounds 5, 9, and 12. Based on the results obtained by cell viability assay, the IC₅₀ values were then evaluated.
Furthermore, to evaluate the selectivity index (SI), the study was also extended to two non-malignant cell lines, HaCaT, a human keratinocyte cell line and 3T3, a mouse embryonic fibroblast cell line. The results obtained clearly demonstrate that, in particular, compounds 9 and 12 exert a marked preferential cytotoxicity toward neuroblastoma cells, while compound 5, although showing a moderate selectivity (SI 1.4–4.8), still exhibits higher cytotoxicity toward malignant cells vs. non-malignant ones. These results indicate a substantial safety window and limited off-target toxicity, even in the absence of lineage-matched healthy controls.
In this regard, it is important to emphasize that, as widely recognized in the neuroblastoma context, there are no physiologically relevant non-malignant sympathoadrenal counterparts for direct comparison. Neuroblastoma originates from transient neural crest-derived sympathoadrenal progenitors, which are present only during embryonic development, and non-malignant cell populations cannot be isolated or maintained as stable in vitro. This biological limitation has prevented the availability of lineage-relevant "healthy" models for neuroblastoma studies.
Nonetheless, the two non-malignant models used here (HaCaT and 3T3) provide significant indications of general cytotoxicity, and the high SI values obtained offer reassuring and promising evidence of selective action against neuroblastoma cells.
- There seems to have some minor typos in the methodology section, consider cleaning them up.
The paper was further checked and typos were corrected.
- Reconcile DPPH “antioxidant” outputs with cellular pro-oxidant effects and avoid causal language without bridging experiments.
In paragraph 3.2, a sentence has been added to try to explain the possible difference between the results of the DDPH assay and the cellular pro-oxidant effects.
- The SwissADME summary is fine for a chemistry-forward paper, it should be supportive, not conclusive.
In the Conclusions was included stating that the predictions obtained with the SwissADME program support the potential of these new compounds and may justify their further development with the aim of obtaining compounds with increased activity.
Reviewer 2 Report
In this report, the authors present design and synthesis of novel coumarin hybrids targeting drug sensitive and resistant neuroblastoma cells. 17 hybrid compounds were designed and synthesized by combining natural antioxidant scaffolds (coumarin, vanillin, and isovanillin) through an acylhydrazone linker. The rationale, synthesis, and biological evaluation are clearly described, and the findings-especially the selective activity of compounds 5, 9, and 12-are significant and well-supported. Overall, the work described valuable insights into hybrid molecule design and shows strong potential for further development.
The manuscript is endowed with sufficient novelty, well-written, and could be accepted for publishing in Antioxidants after minor revision.
- The figures and Schemes appear to have low image resolution; please provide high-quality versions to ensure clarity.
- It would be helpful to indicate the R, R₁, and R₂ substituents in Scheme 2 for the final compounds (1–17), which will enhance structural understanding, readability and SAR analysis. This reviewer believed that instead of representing them (R. R1 and R3) in Figure 1, It should be illustrated in Scheme 2.
Scheme 1 and Scheme 2 should be improve.
Author Response
Reviewer 2
In this report, the authors present design and synthesis of novel coumarin hybrids targeting drug sensitive and resistant neuroblastoma cells. 17 hybrid compounds were designed and synthesized by combining natural antioxidant scaffolds (coumarin, vanillin, and isovanillin) through an acylhydrazone linker. The rationale, synthesis, and biological evaluation are clearly described, and the findings-especially the selective activity of compounds 5, 9, and 12-are significant and well-supported. Overall, the work described valuable insights into hybrid molecule design and shows strong potential for further development.
The manuscript is endowed with sufficient novelty, well-written, and could be accepted for publishing in Antioxidants after minor revision.
We thank the Reviewer for the positive evaluation and we hope that the new experiments conducted will help to further support the promising strong antitumor potential of the hybrid molecules.
- The figures and Schemes appear to have low image resolution; please provide high-quality
versions to ensure clarity.
Figures and diagrams have been inserted, increasing their resolution.
- It would be helpful to indicate the R, R₁, and R₂ substituents in Scheme 2 for the final compounds (1–17), which will enhance structural understanding, readability and SAR analysis. This reviewer believed that instead of representing them (R. R1 and R3) in Figure 1, It should be illustrated in Scheme 2.
Figure 1 has been simplified. The substituents of the various compounds have been inserted in Scheme 2, to improve the structural understanding.
Round 2
Reviewer 1 Report
Issues and concerns raised by the reviewer have been addressed.
Issues and concerns raised by the reviewer have been addressed.
Author Response
Dear Editor,
thank you for appreciating our revised manuscript. As required, we have improved the resolution of Figures 1-2 and Schemes 1-2.
Kind regards
Cinzia Domenicotti and Bruno Tasso
